# Parents’ Willingness and Perception of Children’s Autonomy as Predictors of Greater Independent Mobility to School

**DOI:** 10.3390/ijerph16050732

**Published:** 2019-02-28

**Authors:** Ester Ayllón, Nieves Moyano, Azucena Lozano, María-Jesús Cava

**Affiliations:** 1Faculty of Human Sciences and Education, Department of Psychology and Sociology, University of Zaragoza, Valentín Carderera, 4, 22003 Huesca, Spain; nmoyano@unizar.es; 2Faculty of Human Sciences and Education, Department of Mathematics, University of Zaragoza, Valentín Carderera, 4, 22003 Huesca, Spain; azlozano@unizar.es; 3Faculty of Psychology, Department of Social Psychology, University of Valencia, Avda. Blasco Ibáñez, 21, 46010 Valencia, Spain; Maria.J.Cava@uv.es

**Keywords:** independent mobility, parents, willingness, perception of autonomy, difficulty

## Abstract

The present study aimed to examine the factors associated with different forms of independent mobility (IM) to school (IM one way and IM both ways) according to their parents’ opinions. To do so, several variables were evaluated: how parents assess their children’s autonomy, the difficulty they perceive for IM to school, reasons for IM/no IM to school, parents’ willingness for IM to school, frequency of children’s IM for leisure activities, children having house keys and dangers perceived in the neighborhood. Family-related socio-demographic variables were also assessed: number of children, position occupied by them in the family, family composition, living with both parents or just one, and each parent’s nationality, level of education and job status. This study examined the data collected from 1450 parents (mothers and fathers) with children studying Primary Education years 4, 5 and 6 (*M* age = 10.53, *SD* = 0.90). The results showed that 42.3% of the schoolchildren did not practice IM to school, 18.1% practiced IM one way (they went to or from school alone), and 39.5% practiced IM both way (they went to/from school alone). These findings underline the importance of parents’ willingness for IM to school, and how the balance between how they perceive their children’s autonomy and difficulty for IM is relevant for greater IM to school.

## 1. Introduction

Childhood is a critical development period when physical activity patterns are established, which will probably remain throughout adolescence and adulthood [1,2]. The World Health Organization [3] recommends children being committed to perform at least 1 hour of moderate to vigorous physical activity a day. Children playing, roaming and moving around with no adult supervision naturally promotes physical activity [4]. This independent mobility (IM) aspect facilitates children’s and adolescents’ physical health and psychosocial well-being [5,6,7,8]. IM can contribute to children’s daily physical activity and help them maintain a healthy weight [9]. Moreover, compared to performing structured physical activity (e.g., sporting activities), the possibility of playing and getting around their surroundings without adult supervision improves children’s social interactions and connectedness with friends and other people in their neighborhood. All this implies a major psychosocial benefit for children [9,10]. Certainly, children’s capacity to move around with no adult supervision helps to favor their development at all levels: their physical and mental health, their cognitive performance and, above all, they can establish their socio-emotional relationships and sense of community belonging [11,12,13,14,15].

In line with Hillman et al. and with Carver et al. [16,17], children’s IM can be defined as the skill to play in and get around their surroundings with no adult accompaniment (e.g., going to school, going to nearby parks to play, meeting friends, shopping, catching a bus). This mobility can be considered an indicator of personal independence. According to Brown et al. [18] (p. 386), “independent mobility is a crucial dimension of independence per se”. As children grow, they are given permission to cross roads alone, go to school alone, go to places further away, and need to catch buses and use bicycles to do so. This increasing freedom entails many physic-cognitive and psychosocial benefits [7,9,19]. Moreover, the children who display more IM tend to take some necessary risks for them to develop resilience and to be better prepared to face the adult world [20].

Despite these benefits, compared with previous generations, IM has diminished, particularly in countries like France, Portugal and Italy [21]. According to Prezza et al. [22], seeing children playing or walking in streets, open spaces, parks, etc., in Western countries without strict adult supervision is increasingly more difficult. One of the activities in which children’s IM has drastically reduced is IM to school because increasingly fewer children go to and back from school alone. In the United Kingdom, the percentage of children aged 7–8 years going to school without adult supervision was 80% in 1970, but this percentage was only 10% in 1990 [16]. From 1990 to 2010, the percentage of primary schoolchildren accompanied by an adult on the journey home from school increased in Germany and the United Kingdom [23]. In a recent study conducted in Germany by Scheiner et al. [24], about two thirds of Primary Education schoolchildren are escorted by an adult to school, at least sometimes. In Australia from 1991 to 2012, the percentage of children travelling to school independently was declined from 61% to 32% [25]. This IM to school is, however, positive for them. As the review by Schoeppe et al. [9] indicates, which included 52 studies, most of which focused on IM to and/or from school, IM to school showed significant positive associations with physical activity. From our viewpoint however, no research has analyzed in-depth those children who go independently to or from school, and those who practice both IM forms.

Previous studies [26,27,28] have revealed some spatial, social and cultural changes that have affected the behavior of children: children spend less time playing in cities; the public space in which they can play and socialize has become smaller; the freedom with which they move about has been cut. Moreover, the following reasons all entail less outdoor play: fewer children in today’s society, the few children families have, parents’ and children’s growing safety concerns, and middle-class homes buying more indoor cultural resources [26,28]. The study by Brussoni et al. [29] reflects the importance of encouraging opportunities to play outdoors because if 7–15-year olds play games like “disappearing” or “getting lost”, and engage in risky outdoor play, such leisure allows their physical activity and psychosocial health to increase, while supervised children live more sedentary lifestyles [15,30]. For the above-cited authors, “the word “risk” in the context of risky play denotes a situation whereby a child can recognize and evaluate a challenge, and decide on a course of action” [29] (p. 6425). However, parents’ perceptions of fearing that their children may have a traffic accident and the possible presence of “strangers” have also been associated with restricting their children’s freedom [31,32].

Spending less time playing outdoors means that children in today’s generations are less autonomous. The autonomy concept is defined as a state of being independent or self-governing, and frequently refers to three domains: behavioral, emotional and cognitive [33]. Parents who perceive their children as being more autonomously in cognitive, emotional and behavioral terms could also be more favorable to allow them greater IM. Moreover, parents’ attitude about the importance of enhancing their children’s autonomy can be a relevant variable. Certainly, the family plays a key role in children’s IM. According to the Self-Determination Theory [34,35,36], parents are primary socializers who encourage their children’s autonomy, and are aware that they need this autonomy by taking into account their points of view, and allowing their children to choose, make mistakes and decide how to solve their own problems. In contrast, parents can act by controlling more and leading their children’s conduct according to adults’ criteria. Therefore, parents’ role is fundamental for developing children’s autonomy. This autonomy could be enhanced through behaviors such as allowing their children greater IM. Yet what does parents’ displaying one attitude or another depend on?

Although variables like perceiving danger in social surroundings have been related with children being allowed less IM, other variables like the number of children and the position they occupy in the family, or parents’ willingness to give them more or less IM, can be relevant. However, these variables have been less investigated [37,38]. The present study contemplated examining in-depth the role played by some children’s and parents’ socio-demographic variables, and by the variables related to parents’ perception of the child, in relation with children’s increased IM. Regarding the position with their siblings occupied by children, some studies have demonstrated that those not born first and those who are not an only child enjoy IM somewhat earlier as they are supervised by their brothers and sisters [12,13]. Moreover, older parents allow their children more IM as these parents enjoyed their freedom more than later generations [26]. The fact that both parents work also affects children’s IM. There are increasingly more children whose school hours coincide with their parents’ working hours, and parents take them to school before they go to work [39].

When analyzing parents’ perceptions and their relation with children’s IM, different studies have shown that IM has to do with parents perceiving their neighborhood with a greater sense of community; that is, their neighborhood provides more cohesion, is better connected and safer, and has nearby schools because it belongs to a smaller area [40,41,42,43]. As for parents’ willingness to increase children’s IM, the research by Schoeppe et al. [38] shows how this is also influenced by social cohesion in the neighborhood. When parents perceive this, they are more willing to permit their children cover longer distances for independent travel and outdoor play. The reasons that parents give to either encourage or restrict IM to/from school, and for them performing other leisure activities, have to do with the children’s own characteristics, such as perceiving them to be mature enough and trusting them, the need to protect them or if there are other older children in the family [44,45]. One element which would indicate that parents trust their children is to give them house keys.

Recent studies demonstrate the existence of various forms of IM to school that children enjoy, as well as the importance of distinguishing between children who go to/from school alone (IM both ways) and those who only go to or from school alone (IM one way) [46]. Those children who show IM both ways also report more IM in their leisure activities, and perceive their home as being closer and having fewer difficulties than those who only commute to school one way. These children also perceive their trip to school as being safe, unlike the children who do not practice IM to school. Moreover, predictors for each IM type differ. These findings highlight that IM both ways is differently associated with some IM factors.

By also considering these research works, the present study aimed to examine a considerable number of variables (parents’ and children’s socio-demographic variables, and those relating to how parents perceive their children) in association with IM using a sample formed by the mothers and fathers of children in Primary Education years 4, 5 and 6, and to explore to what extent these variables contribute to predict the IM that parents grant their children. This study specifically analyzed the predictive capacity of the following variables for children’s IM to school: number of children, position the child occupies with their siblings, and parents’ ages, nationality, level of education and job status, parents’ assessment of their children’s degree of autonomy (not autonomous vs. autonomous), parents’ perceived difficulty of IM to school, parents’ willingness for their children to go to school independently, parents’ perception of certain outdoor dangers, parents’ reasons for not allowing IM to school, children’s IM for leisure activities, and children having house keys. These variables were analyzed as possible predictors of different IM forms by distinguishing among no IM, IM one way (only going to/from school) and IM both ways (going to/from school).

## 2. Materials and Methods

### 2.1. Study Sample and Design

Letters for participation were sent to 12 schools from Huesca (Spain) to invite them to participate in the study. The research team contacted each school to provide information about the study and the procedure. Eleven of the 12 schools volunteered to take part. Two researchers went to these schools and handed out paper-and-pencil questionnaires to children. Children handed each questionnaire to their parents with a letter providing instructions to complete it. An approximate 10-day deadline date was set to return questionnaires. Children handed questionnaires to their corresponding teacher. Researchers collected questionnaires from schools.

Data were collected from 1835 mothers and fathers of the schoolchildren in Primary Education years 4, 5 and 6, which usually covers ages from 9 to 11 years old, from the 11 participating education centers (seven were public, four were semi-private) from Huesca (Spain) between February and May 2017. As some families’ place of residence was not in the city of Huesca, 191 cases were removed as they lived in rural areas or did not properly complete the required information. Indeed no families living further than 3.5 km from school were included, and 194 subjects were not considered as they referred to only one parent. Thus the final sample was formed by 1450 parents. Of their questionnaires, 725 were completed by mothers, 718 by fathers, and 7 by the mother’s partner. About 46.8% of their children were boys and 53.2% were girls. The children’s mean age was 10.53 (*SD* = 0.90) within a range from 8.16 to 12.88. Only four children were 8 years old and 86 were 12 years old. This study forms part of a larger European project called CAPAS-Ciudad (BLINDED).

### 2.2. Materials

Socio-demographic background. It included questions about family composition or co-existence at home, where living with both parents or only one was distinguished. Data about both mothers and fathers was obtained (age, nationality, level of education and job status).

The questionnaire filled out by parents was “The Independent Mobility of Italian children” (L’autonomia di movimento dei bambini italiani) by Francesco Tonucci et al. [47], translated into Spanish with these authors’ consent. This questionnaire has been taken into account in other studies [22,47], and allows the following variables to be evaluated:

IM to/from school. Parents completed a self-reported measure with the following questions: *Does your child go to school by him/herself?* and *Does your child come back from school by him/herself?* For both questions, the answer options were dichotomous: Yes or No. A dummy variable was created to distinguish: IM to or from school (IM one-way), IM to and from school (IM both ways) and no IM, which scored 1, 2 and 0, respectively.

Evaluating your child’s autonomy. A question was included that indicated: *What is your child’s attitude toward you?*, for which there were two options: not autonomous vs. autonomous.

Perceived difficulty for IM to school. They were asked *Is it difficult for your child to go to school by him/herself?* Parents had to answer “*Yes*” or “*No*” (2 and 1, respectively).

Reasons for not allowing IM to school. Parents were asked to mark whether: “*We do not feel able to let him/her going by him/herself*” and “*Our child does not want to go by him/herself*”. Some other reasons were also provided, such as: *I prefer spending this time with my child*; *my child is too young*; *my child could have bad experiences*.

Parents’ willingness for IM to school. Whether parents would have liked their children to go to school without adult accompaniment was measured by the response answers “*Yes*” or “*No*”.

Frequency of IM for outdoor leisure activities. Besides IM to school, it was considered important to assess the frequency by which schoolchildren performed certain outdoor leisure activities unsupervised by adults. Parents were asked *How frequently does your child do this activity without being accompanied by an adult*? and the activities were: 1) *visiting friends and out-of-school activities*; 2) *using public transport*; 3) *cycling around the neighborhood*; 4) *buying from a shop*; 5) *playing outside (in a park or open spaces)*; 6) *going out when it gets dark*. All these options were answered on a 4-point frequency scale: “*Never*”, “*Sometimes*”, “*Most of the time*” and “*Always*”.

Having house keys. Parents answered the following question: How often does your children have house keys?, with answer options ranging from: *Never* (1) to *Always* (4).

Perceived dangers. The assessed barriers included danger related to streets, footpaths and crossing roads, public parks and entertainment areas, buses and public transport, and shops or supermarkets. All the items were assessed on a 4-point Likert scale ranging from (1) *strongly disagree to* (4) *strongly agree*.

## 3. Data Analysis

First of all, the sample’s socio-demographic characteristics were analyzed. A comparative analysis was done by performing a MANOVA of the variables evaluated for mothers and fathers. As no significant differences were found, further control of parents’ gender was not considered. We next examined the percentage of parents who indicated that their children went to school with no adult accompanying them (IM to school) by distinguishing: who went to school accompanied by adults (no IM); who went to or returned from school alone (IM one way); who went to and returned from school alone (IM both ways). At this point, we also examined whether there were differences in IM based on children’s gender and school year.

We analyzed if there were any significant differences in children’s IM level by considering the socio-demographic and parents’ perception variables. To do so, we conducted a MANOVA to analyze any possible differences among the three groups with different levels of children’s IM (no IM, IM one way, IM both way) in the socio-demographic variables (children’s school year, number of children, their position in the family, living with either both parents or only one, and parents’ age, nationality, level of education and job status) and the variables related to parents’ perceptions and attitudes (parents’ perceptions about their children’s autonomy, perceived difficulty of IM to school, the reasons for not allowing IM to school, parent’s willingness for their children’s IM to school, giving their child house keys, frequency of IM for other leisure activities and perceived dangers). In these analyses, the categorical variables, which were mostly dichotomous, were treated as continuous variables. As there were three groups with different levels of children’s IM, post hoc tests were run to explore the significant differences between means.

Finally, in order to analyze the predictive value of the examined variables (socio-demographic variables and the variables related to parents’ perceptions and attitudes) on children’s IM, we conducted several binary logistic regression analyses, in which a dependent variable was established in each case as follows: a) no IM vs. IM one-way (the dichotomous variable was 0 and 1, respectively) and b) no IM vs. IM both ways (the dichotomous variable was 0 and 2, respectively). The socio-demographic variables and variables related to parents’ perceptions and attitudes were included in the regression analyses as predictor (independent) variables.

## 4. Results

The sample’s socio-demographic characteristics are presented in Table 1. The mean age of mothers and fathers was 43.02 (*SD* = 5.03) and 45.38 (*SD* = 5.43), respectively. Their nationality was mainly Spanish, and their level of education and job status were high. The chi-squared test found significant differences between both parents for all the evaluated variables, except for employment. Compared to fathers, the surveyed mothers were significantly younger, the nationality of more of them was not Spanish and they had a higher level of education.

Later, the percentages of the parents who indicated that their children went to school accompanied by adults (no IM), went to or returned from school alone (IM one way), and went to/from school alone (IM both ways) were examined. Our results indicated that 42.3% of parents stated that their child went accompanied by an adult to school (no IM), 18.1% reported that their child went to school or returned from school alone (IM one way), and 39.5% pointed out that their child went to/from school alone (IM both way). Regarding children’s gender, no significant differences between boys and girls in their IM levels were found (F(1, 1352) = 0.312, *p* = 0.517). However, significant differences in children’s IM according to school year were observed (F(2, 1349) = 91.75, *p* < 0.001) for all three school years. Greater IM was found for children studying in Primary Education year 6. Therefore, we included this variable in subsequent analyses.

As Table 2 shows, significant differences were found according to the IM group in the socio-demographic variables: school year, number of children, children’s position in the family, both parents’ age and nationality and mothers’ job status. Specifically, the IM both ways group included more children studying in Primary Education year 6, more children in the family, and children occupying a more intermediate or advanced position with their siblings compared to the no IM and IM one way groups. In the groups with more IM (IM one way and IM both ways), both parents were older than those in the no IM group, and the IM one way and IM both ways groups were made up of more parents with Spanish nationality. The IM one way group was formed by more working mothers compared to the no IM and IM both ways groups. However, neither parents’ level of education nor fathers’ job status was associated differently with each IM group.

Moreover, differences were observed with the other evaluated variables, which were significant in the perceived difficulty of IM to school and for all the reasons for IM according to each group (see Table 2). The no IM group was formed by more parents perceiving greater difficulty for their children’s IM, followed by the IM one way and the IM both ways groups. The no IM group obtained the highest score for this variable, as did those parents who informed they were feeling less willing to allow their children to go to school alone. The reasons reported in this group included their child being too young, unlike the other two groups. However, the IM both ways group informed that if their children were not allowed to go to school alone, it was because parents preferred spending this time with their child or they wished to avoid their child having unpleasant experiences on their way to school. Significant differences were also observed in IM in situations related to outdoor leisure activities, having house keys and some perceived dangers. In general, the IM both ways group reported their children’s more frequent IM for outdoor leisure activities, and more of them had house keys. Regarding perceived dangers, the no IM group obtained higher scores for infrastructure-related dangers, such as streets, footpaths and crossings, and for specific spaces like shops and supermarkets.

As seen in Table 3, the predictor variables for IM one way were school year, frequency with which the children performed other outdoor leisure activities independently, and having house keys. Together, these variables explained 24% of the variance for IM one way.

Table 4 shows for the prediction of IM both ways versus no IM the following predictor variables: the child occupying a more advanced position with his/her siblings, more frequency of IM for certain outdoor leisure activities, and having house keys. However, the fact that parents perceived more difficulty for IM to school was associated with less IM. Parents’ willingness for their children’s IM to school and children’s autonomy perceived by their parents were positively related with IM both ways. This model explained 55% of variance.

## 5. Discussion

The objective of the present study was to assess those factors associated with different forms of IM to school in schoolchildren studying Primary Education years 4, 5 and 6 (corresponding mainly to the ages of 9–11 years) by considering their parents’ opinions. Of all the participating schoolchildren, 42.3% practiced no IM, 18.1% practiced IM one way, and 39.5% practiced IM both ways. It is noteworthy that over half our evaluated sample went to/from school alone, with most of them doing so both ways. This prevalence is consistent with that found in previous studies conducted in other Spanish cities [48], even though the above-cited study only took into account the journey made to school in the morning from children’s homes. The percentage of children’s IM observed herein (including IM both ways and IM one way) was slightly lower to that found in Germany and Canada [24,49] for children of similar ages. In line with previous studies [7,8,21,24,49], children’s age (school year) was a variable related to IM and older children show greater IM to school. However, other variables were also found to be relevant in relation to children’s IM to school.

The present study shows that those parents who allowed more IM (the IM both ways group) were those with a given family composition as they had more children and the studied children occupied a more intermediate or advanced position with their siblings. As other studies have observed, the children who are neither an only nor first-born child, and can be accompanied by their siblings, have more freedom to move around [12,13]. Moreover, older parents of Spanish nationality probably come from a generation who enjoyed more IM in the place where they lived. As indicated by Karsten [26], a generation-related change has taken place in the use of urban spaces. This author also points out that there are “indoor children” as they ‘play’ indoors (i.e., mainly watching TV), they do not participate in many other activities”, and many of these children are of immigrant origin ([26], p. 285). The parents of Spanish nationality seemed to perceive less difficulty for their children’s IM than the parents of immigrant origin, which is an important factor that restricts children’s IM: parents perceiving possible difficulties or dangers related to traffic and undesirable people [42,50]. Similarly, the results of this study showed that the parents in the no IM group perceived more dangers in streets, on footpaths and crossings, and in shops and supermarkets. A recent study [28] showed how parents reduced their children’s IM and playing outdoors because they were worried about their safety, which was also related with their level of socio-economic hardships. However, they also perceived their own childhood as being better, and they acknowledged that they should maximize their child’s independent outdoor play and limit them using screens on electronic devices.

Differences were also found in the reasons explaining IM because the reason why the parents in the no IM group did not like their child going to school alone was because “my child is too young”. However, the main reason for the IM both ways group was “I prefer to spend more time with my child” or “to avoid possible bad experiences”; that is, the parents who did not encourage IM felt that their child was not mature enough and uncapable of IM [44,45]. In line with other previous studies, the present research results found that the children who practiced more IM to school also had more IM to perform other outdoor leisure activities [22,45]. Thus it could be deduced that children’s freedom to move around increased according to the opportunities they had to do so.

Different factors were found when analyzing the significant correlates of IM for its categories (i.e., IM one way, IM both ways, and no IM). School year, having more IM for activities like visiting friends, out-of-school activities and going to shops to buy, along with having house keys, were the main predictors of IM to school one way vs. no IM. Regarding school year, one explanation for this finding could be that each school year frames certain changes and independency for children. Indeed it is likely that parents decide to provide their children with more independency to change certain habits at the beginning of the school year than during it; moreover with each school year, children are older and parents could decide to give them more IM to school. The present study also showed that the children with greater autonomy to visit friends, perform out-of-school activities, buy in shops and have house keys also performed greater IM to school. Compared with the IM one way predictors, the IM both way predictor variables were related more with aspects like barely perceiving difficulty, more parents’ willingness for IM and perceiving the children as autonomous, which underlines the importance of distinguishing between both types of IM. One main contribution made by this study is that it showed how the parents who allowed their children more IM (both ways) were also willing for their children to go to school alone. These parents also perceived fewer difficulties in their neighborhood, valued their child as more autonomous, and they possibly perceived that promoting their children’s autonomy was positive for their psychosocial development to a greater extent. In line with this, it would be convenient to analyze the attitudes that parents have about the importance of promoting their children’s autonomy in future studies. In relation to this, a recent study [51] identified three different parent "profiles", namely promotors, pragmatic and protectors, for their child’s IM depending on the attitudes shown by parents. In other words, there were parents who encouraged their child’s autonomy or were more controlling and protecting, and they addressed their child’s behavior according to their adult criteria [35]. The social setting was also found to have an influence as these attitudes were clearly more influenced by social rules and the neighborhood as all this forms part of a major social network. Social rules mean that being “good parents” sometimes implies restricting IM [52], whereas close supervision is taken as a typical characteristic of good care [53]. Regarding one’s neighborhood, Schoeppe et al. [38] showed that those parents with high perceptions of good social cohesion in their neighborhood were more willing to allow longer IM distances (to get around without adult supervision and to play in public spaces). Conversely, the study by Villanueva et al. [54], which was conducted in Australia with 10- to 12-year-old boys and girls, indicated greater IM when both children and their parents trusted that they were capable of IM. Here the authors concluded that it was fundamental to provide children with strategies to get around safely (e.g., estimated distance, the direction taken, spatial reference skills, etc.) to be able to enhance IM and to provide less supervision.

Another variable that predicted IM to school both ways herein was a higher frequency IM for visiting friends or performing out-of-school activities. The literature reveals how children with more independence to take a route to school also display more independence to go and perform their out-of-school activities, and to use public spaces in general [18,22,55]. Having house keys is an indicator related with the other variables that predicted IM both ways because, when parents decide to give their children house keys, they do so because they perceive their child as being responsible enough to receive this sign of autonomy (and they value this attitude). So they trust their child and their neighborhood. Hence they perceive fewer difficulties and show more willingness to encourage IM to school. Finally, the position that the child occupies among siblings also predicts IM to school both ways. In our conceptualization of IM in line with Bhosale [4], we considered that the child was still independent even if accompanied by his/her siblings.

One main aspect of the present study was the high participation rate of the schools in the city of Huesca (11 of 12 schools; 92%). Moreover, one highlighted contribution of this study is having analyzed three different forms of IM (no IM, IM one way and IM both ways). Unlike other research works, which have analyzed only IM one way, our study explored many variables associated with IM both ways, a type of IM that implies children’s greater independence. The present research results help to provide a better understanding of the variables that influence children’s IM and could be very useful for developing intervention proposals to encourage it. Certainly, parents are fundamental for their children’s greater IM, and the results offered herein allow us to advance in our knowledge about which family variables (both socio-demographic variables and parents’ perceptions) are related more with IM both ways. Indeed the results obtained herein stress that IM both ways is associated with three interlinking aspects linked with parents’ perceptions and attitudes, namely: greater parents’ willingness for their children to acquire IM, parents perceiving fewer difficulties in their neighborhood, and them evaluating their child as being more autonomous. It would be worthwhile making these variables the priority objective in interventions made with families to favor their children’s IM to school. Some parents’ fears, or the difficulty they perceive with their social setting, may also require making more marked changes in the way cities and communities are organized. In other cases, it might be necessary to revise with families some social rules about conducts associated with being a “good mother”/“good father” as some are possibly linked with their willingness, or not, for their children’s IM to school. It would also be worthwhile analyzing with parents the specific difficulties they have with their children’s IM both ways, what changes should be made in their neighborhood to reduce these difficulties, and to explore with parents what degree of autonomy they actually perceive of their children and how it can be increased by allowing them to see that, as their children grow, their autonomy must increase to favor their better psychosocial development.

In the social context in which this research was conducted, we believe that work is being done, and should continue to reach a public policy, to help reduce the perception of difficulty in a neighborhood (improved signposting, more policemen/women participating, more social participation by shop owners to offer children of contacting their parents when they face possible difficulties, parents themselves participating on corners in cities (encounter points), schools participating in IM to school, etc.). We also believe that we have to empower children, which is important to allow them to speak so they may participate in how they wish to develop their daily IM [56]. IM can be encouraged by facilitating socially enriching experiences like the possibility of getting around the city with siblings, peers and pets [57]. All this would doubtlessly entail improved quality of life for all citizens. In addition, cultural aspects linked to parental socialization styles and social norms should also be considered in future research. In this line, previous studies [58,59] have shown that cultural differences are related to differences in parental childrearing strategies and parental socialization goals. Taverna et al. [58] observed differences between German and Italian cultures in childrearing practices, and the levels of autonomy and independence that mothers expect in their children.

This study has its limitations. First, our results cannot be generalized to other urban or rural areas in Spain, or abroad, as our participants came from one particular city in Spain. Thus future research works should be extended and conducted in other cities of Spain, and should longitudinally bear in mind the perspectives of both children and their parents at the same time, and how they influence IM. Two-way and dynamic influences may exist between the evaluations made by parents and children, which might promote or undermine a children’s autonomy from developing. So both parents’ and children’s perceptions of IM in the same research as part of future studies would be worthwhile, along with their reasons for IM both ways to school.

## 6. Conclusions

In recent decades, children’s IM to school has substantially reduced, which implies negative consequences for their health and psychosocial well-being. This study has analyzed several family variables associated with different forms (IM one way, IM both ways) of children’s IM to school, and provides interesting and novel data about these variables. The obtained results reveal that the families which permit more IM (both ways) are those that wish their children to go to school alone, they perceive fewer difficulties in their social setting, and consider their children more autonomous. The results of this study have highlighted the importance of parents’ willingness for greater IM, thus it would be convenient to continue exploring parents’ reasons for this willingness and to what extent their perception of the importance of enhancing children’s autonomy may be related with allowing their children’s IM to school. One of the main contributions of this study is the importance of the parents’ willingness for children’s IM to school, and future studies should examine this variable and what factors influence it in in-depth.

Taking into account the importance of parental perceptions, it would also be advisable to develop interventions to make parents aware of the benefits that children’s IM to the school has for the health and psychosocial well-being of their children. Although some environmental and organizational aspects of cities (e.g., accessible streets, fewer cars, distance to school, etc.) should be changed to favor children’s IM to school, parental perception of the possible dangers and difficulties for children’s IM, and probably what they think about the behaviors that are also socially expected of a “good father”/“good mother”, could also be key variables to help increase children’s IM to school in our societies.

## Figures and Tables

**Table 1 ijerph-16-00732-t001:** The sample’s evaluated socio-demographic characteristics (*N* = 1450).

Variables	Mothers	Fathers	
	*n* (%)/*M(SD)*	*n* (%)/*M(SD)*	χ^2^/F
	725 (50)	725 (50)	
**Age**	43.02 (5.03)	45.38 (5.43)	5.040 ***
**Nationality** Foreign Spanish	89 (12.3) 633 (87.2)	58 (8) 663 (92)	635.57 ***
**Education** None Primary-secondary High school Professional training University degree	11 (1.5) 107 (14.7) 102 (14.1) 108 (14.9) 393 (54.1)	15 (2.1) 155 (21.4) 130 (18) 159 (22.1) 262 (36.3)	1106.07 ***
**Employment** Yes No	609 (84) 113 (15.6)	686 (94.5) 31 (4.3)	2.41

*** *p* < 0.01.

**Table 2 ijerph-16-00732-t002:** Descriptive statistics of the examined variables and significant differences based on MANOVA.

Variables	No IM	IM one way	IM both ways	χ^2^/F
	*n (%)* *M(SD)*	*n (%)* *M(SD)*	*n (%)* *M(SD)*	
**Socio-demographic variables**				
School year 4-year (9-year-old children) 5-year (10-year-old children) 6-year (11-year-old children)	295 (59.2%) 195 (41.5%) 71 (18.5%)	80 (16.1%) 98 (20.9%) 76 (19.8%)	123 (24.7%) 177 (37.7%) 237 (61.7%)	91.75***
No. of children in family	1.99 (0.91)_a_	1.97 (0.64)_a_	2.15 (0.89)_b_	6.20**
Position (with siblings)	1.42 (0.73)_a_	1.41 (0.60)_a_	1.75 (0.82)_b_	32.57***
Living with (0 = one parent, 1 = both parents)	0.91 (0.27)	0.94 (0.22)	0.93 (0.24)	1.10
Mothers’ age	42.37 (4.58)_a_	43.85 (1.10)_b_	43.33 (1.19)_b_	9.86***
Mothers’ nationality (1 = not Spanish, 2 = Spanish)	1.90 (0.29)_b_	1.89 (0.30)_b_	1.84 (0.36)a	5.40**
Mothers’ education (0 = none to 5 = university degree)	4.04 (1.16)	4.27 (1.14)	4.01 (1.21)	4.64
Mothers’ employment (1 = employed, 2 = unemployed)	1.16 (0.37)_b_	1.09 (0.28)_a_	1.17 (0.37)_b_	5.80**
Fathers’ age	44.39 (5.15)_a_	46.10 (5.05)_b_	46.11 (5.40)_b_	14.96***
Fathers’ nationality (1 = not Spanish, 2 = Spanish)	1.94 (0.22)_b_	1.91 (0.27)	1.89 (0.30)_a_	4.70**
Fathers’ education (0 = none to 5 = university degree)	3.70 (1.19)	3.83 (1.19)	3.64 (1.25)	2.19
Fathers’ employment (1 = employed, 2 = unemployed)	1.04 (0.18)	1.05 (0.21)	1.05 (0.21)	0.06
**Evaluating your child’s autonomy** (1 = not autonomous, 2 = autonomous)	1.50 (0.50)_a_	1.64 (0.53)_b_	1.76 (0.42)_c_	39.86***
**Perceived difficulty for IM to school** (1 = no, 2 = yes)	1.47 (0.53)_c_	1.18 (0.49)_b_	1.04 (0.26)_a_	142.84***
**Reasons for not allowing IM to school**				
We do not feel able to let him/her go by him/herself (1) Our children does not want to go by him/herself (2) (range = 1–2)	1.21 (0.49)_a_	1.39 (0.55)_b_	1.28 (0.47)	7.68***
**Other reasons for not allowing IM to school** (0 = no, 1 = yes)				
I prefer spending this time with my child	0.27 (0.44)_a_	0.39 (0.49)	0.40 (0.49)_b_	9.79**
My child is too young	0.43 (0.49)_b_	0.21 (0.41)_b_	0.10 (0.30)_a_	51.90***
My child could have bad experiences	0.27 (0.44)_a_	0.30 (0.46)_a_	0.47 (0.50)_b_	16.29***
**Parents’ willingness for IM to school** (1 = no, 2 = yes)	1.55 (0.49)_a_	1.75 (0.43)_b_	1.87 (0.33)_c_	49.09***
**Frequency of IM for outdoor leisure activities** (range 1 = never to 4 = always)				
Visiting friends, out-of-school activities	1.30 (0.58)_a_	1.82 (0.75)_b_	2.28 (0.92)_c_	156.78***
Using public transport	1.02 (0.12)	1.06 (0.39)	1.07 (0.36)	3.42
Cycling around the neighborhood	1.10 (0.33)_a_	1.22 (0.54)_b_	1.31 (0.64)_b_	17.08***
Buying from a shop	1.58 (0.62)_a_	1.98 (0.55)_b_	2.08 (0.62)_b_	67.39***
Playing outside, in a park or open spaces	1.39 (0.59)_a_	1.70 (0.68)_b_	2.03 (0.79)_c_	82.79***
Going out when it is gets dark	1.02 (0.15)_a_	1.03 (0.17)_a_	1.09 (0.29)_b_	8.99***
**Having house keys** (range of frequency 1 = never to 4 = always)	1.29 (0.78) _a_	1.96 (1.27)_b_	2.56 (1.32)_c_	124.29***
**Perceived dangers** (range 1 = not at all dangerous, 4 = very dangerous)				
Street, footpaths, crossing roads	2.54 (0.77)_c_	2.39 (0.75)_b_	2.20 (0.60)_a_	19.23***
Public parks, entertainment areas	2.08 (0.69)	1.98 (0.68)	1.99 (0.62)	2.42
Buses and public transport	2.29 (0.75)	2.18 (0.72)	2.25 (0.80)	1.37
Shops and supermarkets	2.05 (0.71)_b_	1.88 (0.66)_a_	1.89 (0.61)_a_	7.49**

*Notes*: Different subscripts indicate significant differences between means (*p* < 0.05): a < b < c. ** *p* < 0.05; *** *p* < 0.01

**Table 3 ijerph-16-00732-t003:** Logistic regression analysis for predicting of no IM vs. IM one way.

Variables	B	Se	*p*-Value	Exp(B)
School year	0.30	0.12	0.011	1.35
Frequency of IM for visiting friends, out-of-school activities	0.78	0.12	0.000	2.20
Frequency of IM for shopping	0.32	0.14	0.026	1.38
Having house keys	0.40	0.08	0.000	1.50

**Table 4 ijerph-16-00732-t004:** Logistic regression analysis for predicting of no IM vs. IM both ways.

Variables	B	Se	*p*-Value	Exp(B)
Position (with siblings)	0.39	0.14	0.004	1.49
Frequency of IM for visiting friends, out-of-school activities	1.02	0.15	0.000	2.79
Having house keys	0.69	0.10	0.000	1.99
Perceived difficulty for IM to school	−1.53	0.35	0.000	0.28
Parental willingness for IM to school	1.77	0.38	0.000	5.91
Evaluating your child’s autonomy	0.79	0.23	0.001	2.22

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
