# Peer review of "Parents’ Willingness and Perception of Children’s Autonomy as Predictors of Greater Independent Mobility to School"

_ijerph, 2019, doi:10.3390/ijerph16050732_

Round 1
Reviewer 1 Report
The present paper aimed to examine the factors associated with different forms (one way
and both ways) of children’s independent mobility (IM) to and from school according to their parents’ opinions.
One of the most important contributions of the survey concerns the emphasis on family‐related socio‐demographic variables, in particular: number of children, position occupied by them, family composition, living with both parents or just one, parent's nationality, level of education and job.
These variables, that are crucial for independent mobility, have not been studied in depth in previous investigations, in particular, the role of family variables.
Another important contribution of the article concern the emphasis of the survey on parents’ assessment of their child’s autonomy, parental willingness for IM to school, and dangers perceived in the neighbourhood; findings especially underline the importance of parental willingness as to their children’s travelling to school alone.
The introduction provides sufficient background as to the importance of fostering children's independent mobility and of identifying the factors that foster or hinder it.
The effects of independent mobility on physical health and on the psychosocial well-being of children are also highlighted.
It would be interesting to expand the part describing the benefits of independent mobility on children's psychosocial health, and to describe in greater detail the health benefits of unstructured physical activity (free play and independent mobility) compared to those produced by structured physical activity (e.g., sporting activities).
Moreover, it would be interesting to insert a part concerning international data on independent mobility in order to contextualize the data which emerged from the survey.
The research design is appropriate because the survey only focuses on parents' opinions regarding independent mobility and takes into account many, both socio-demographic and family, variables; for this reason the research involved only parents.
It would be helpful to explain in more detail why the authors chose to administer the questionnaires only to parents, and not to children as well.
In a subsequent study, as also indicated by the authors, it would be interesting to administer a questionnaire to children, in order to assess if and how the responses of children and parents vary and in relation to which variables. It would also be interesting to take into consideration different geographical areas in Spain, and see if and how this geographical variable affects children’s independent mobility and parents’ opinions.
Methods are described appropriately, in addition, the investigated variables are clearly indicated.
In a subsequent study, it would be interesting to evaluate parents’ sense of community, through the use of internationally validated scales, given authors’ emphasis on the perception of danger and on the importance of the perception of the sense of security of the neighborhood.
The analysis of results might be ameliorated. In order to do so, it would be useful to insert, before presenting the overall results, a synthetic part describing the main statistical analyzes performed, highlighting the results obtained for each analysis.
The discussion of results should focus only on the particularly relevant and innovative aspects that emerged from the survey; at the same time, the conclusions should be expanded, giving more emphasis and scientific support to the innovative and relevant aspects that emerged.
The questionnaire given to parents in the research was not used in the two research studies indicated in the References (14, 40).
Author Response
RESPONSES TO REVIEWER 1
The present paper aimed to examine the factors associated with different forms (one way and both ways) of children’s independent mobility (IM) to and from school according to their parents’ opinions.
One of the most important contributions of the survey concerns the emphasis on family related socio demographic variables, in particular: number of children, position occupied by them, family composition, living with both parents or just one, parent's nationality, level of education and job.
These variables, that are crucial for independent mobility, have not been studied in depth in previous investigations, in particular, the role of family variables.
Another important contribution of the article concern the emphasis of the survey on parents’ assessment of their child’s autonomy, parental willingness for IM to school, and dangers perceived in the neighbourhood; findings especially underline the importance of parental willingness as to their children’s travelling to school alone.
The introduction provides sufficient background as to the importance of fostering children's independent mobility and of identifying the factors that foster or hinder it.
The effects of independent mobility on physical health and on the psychosocial well-being of children are also highlighted.
First of all, we would like to thank the reviewer for these comments. We also thank him/her for all the provided feedback as we believe that responding to the revision has improved the original version of the manuscript. We appreciate the constructive and thoughtful feedback and suggestions. We systematically describe the changes made to address the reviewer's suggestions, and we provide an account of each response.
It would be interesting to expand the part describing the benefits of independent mobility on children's psychosocial health, and to describe in greater detail the health benefits of unstructured physical activity (free play and independent mobility) compared to those produced by structured physical activity (e.g., sporting activities).
Answer: Following this suggestion, we have extended the description of the benefits of independent mobility on children’s psychosocial health, and we have compared these benefits to those produced by structured physical activity.
Lines 38-46: IM can contribute to children's daily physical activity and help them to maintain a healthy weight [9]. Moreover, compared to performing structured physical activity (e.g., sporting activities), the possibility of playing and getting around their surroundings without adult supervision improves children’s social interactions and connectedness with friends and other people in their neighborhood. All this implies a major psychosocial benefit for children [9,10]. Certainly, children’s capacity to move about with no adult supervision helps to favor their development at all levels: their physical and mental health, their cognitive performance and, above all, they can establish their socio-emotional relationships and sense of community belonging [11-15].
Moreover, it would be interesting to insert a part concerning international data on independent mobility in order to contextualize the data which emerged from the survey.
Answer: We have inserted a part about international data on independent mobility into the revised manuscript.
Lines 60-67: In the United Kingdom, the percentage of children aged 7-8 years going to school without adult supervision was 80% in 1970, but this percentage was only 10% in 1990 [16]. From 1990 to 2010, the percentage of primary schoolchildren accompanied by an adult on the journey home from school increased in Germany and the United Kingdom [23]. In a recent study conducted in Germany by Scheiner et al. [24], about two thirds of Primary Education schoolchildren are escorted by an adult to school, at least sometimes. In Australia from 1991 to 2012, the percentage of children travelling to school independently was declined from 61% to 32% [25].
Lines 297-299: The percentage of children’s IM observed in this study (including both ways and one way) is slightly lower to that found in Germany and Canada [24,49] for children of similar ages.
The research design is appropriate because the survey only focuses on parents' opinions regarding independent mobility and takes into account many, both socio-demographic and family, variables; for this reason, the research involved only parents.
It would be helpful to explain in more detail why the authors chose to administer the questionnaires only to parents, and not to children as well.
In a subsequent study, as also indicated by the authors, it would be interesting to administer a questionnaire to children, in order to assess if and how the responses of children and parents vary and in relation to which variables.
Answer: We agree that parents and children’s opinions would have enriched this study. However, in this study, we were interested in obtaining parents’ in-depth opinions. We thank the reviewer for this suggestion and we would certainly like to conduct a more detailed future study in which both aspects would be considered together. Thank you for this suggestion.
It would also be interesting to take into consideration different geographical areas in Spain, and see if and how this geographical variable affects children’s independent mobility and parents’ opinions.
Answer: We thank the reviewer for this suggestion. We will consider it for further research so as to extend the present findings.
Methods are described appropriately, in addition, the investigated variables are clearly indicated.
Answer: We thank the reviewer for this comment.
In a subsequent study, it would be interesting to evaluate parents’ sense of community, through the use of internationally validated scales, given authors’ emphasis on the perception of danger and on the importance of the perception of the sense of security of the neighborhood.
Answer: We thank the reviewer for this suggestion. Sense of community, identity as part of the community and the way we rely on others are undoubtedly interesting variables that would enrich our understanding of how the social network really outlines the way we behave and to extend children´s freedom.
The analysis of results might be ameliorated. In order to do so, it would be useful to insert, before presenting the overall results, a synthetic part describing the main statistical analyzes performed, highlighting the results obtained for each analysis.
Answer: Following this suggestion, we have provided a “Data analysis” section prior to the “Results” in the revised manuscript (see lines 202-229).
The discussion of results should focus only on the particularly relevant and innovative aspects that emerged from the survey; at the same time, the conclusions should be expanded, giving more emphasis and scientific support to the innovative and relevant aspects that emerged.
Answer: Following this reviewer’s suggestion, we have revised the discussion of the results and have also extended the conclusions by placing more emphasis and providing scientific support to the innovative and relevant aspects that emerge.
Lines 422-441: In recent decades, children’s IM to school has substantially reduced, which implies negative consequences for their health and psychosocial well-being. This study has analyzed several family variables associated with different forms (IM one way, IM both ways) of children’s IM to school, and provides interesting and novel data about these variables. The obtained results reveal that the families which permit more IM (both ways) are those that wish their children to go to school alone, they perceive fewer difficulties in their social setting, and consider their children more autonomous. The results of this study have highlighted the importance of parents’ willingness for greater IM, thus it would be convenient to continue exploring parents' reasons for this willingness and to what extent their perception of the importance of enhancing children’s autonomy may be related with allowing their children’s IM to school. One of the main contributions of this study is the importance of the parents’ willingness for children’s IM to school, and future studies should examine this variable and what factors influence it in in-depth.
Taking into account the importance of parental perceptions, it would also be advisable to develop interventions to make parents aware of the benefits that children’s IM to the school has for the health and psychosocial well-being of their children. Although some environmental and organizational aspects of cities (e.g., accessible streets, fewer cars, distance to school, etc.) should be changed to favor children’s IM to school, parental perception of the possible dangers and difficulties for children’s IM, and probably what they think about the behaviors that are also socially expected of a “good father”/ “good mother”, could also be key variables to help increase children’s IM to school in our societies.
The questionnaire given to parents in the research was not used in the two research studies indicated in the References (14, 40).
Answer: We thank the reviewer who detected this mistake, which has been corrected. The corresponding references for the questionnaires were those with these numbers: 22 and 47, respectively.
Finally, we would like to thank this reviewer for all the comments because they have helped us to improve our manuscript for publication in the International Journal of Environmental Research and Public Health.

Reviewer 2 Report
Review of ijerph-430852
Parental Willingness and Children’s Autonomy Perception as Predictor Factors of Greater Independent Mobility to School
Thank you for the opportunity to review this manuscript. The study described addresses an interesting and worthwhile topic – parental factors associated with different forms of children’s independent mobility to school. The study is well conceptualized and provides an in-depth view on the parental perspective of children’s autonomy.
In broad terms, the manuscript conforms to an appropriate structure and style, but at a more detailed level, it needs some work. There are a few matters of general grammatical construction and vocabulary. Furthermore, the use of different variants of essentially the same term - which may be deliberate and intended to make the narrative more varied - in my view just leads to more work for the reader in cross-referencing and confirming interpretations. It would be better to decide on a specific term and then using it consistently. There are also some issues regarding methodological explanations and analysis. Below I have made detailed suggestions for improvements.
A major methodological concern is the missing inclusion of important child’s characteristics in the analysis (only number of children and position). Many studies evaluated that a child’s gender and age hardly influence the extent of independent mobility parents granted their child (Buliung, Larsen, Faulkner, & Ross, 2017; Pacilli, Giovannelli, Prezza, & Augimeri, 2013; Scheiner, Huber, & Lohmüller, 2019; Shaw et al., 2015). For example, Scheiner et al. (2019) found that the feeling that the child is too young is mainly associated with children younger than eight years.
Including only parents characteristics and opinions and including parents of children from three different school years (4-6) in the sample, can lead to a misinterpretation of the results, e.g. if all children of the “no autonomy” group are within the same age. For this reason, if age and gender of the child is available, the data should be analyzed again be separating age/gender groups or controlling for child’s age/gender.
Please add a section describing the applied statistical analysis of the study as last chapter of “Materials and Methods” to provide the opportunity of reproducibility. (after line 187)
In the conclusion (line 381) it is mentioned that the family value the encouragement of autonomy for their child’s development. However, no method or result is described which evaluated the importance of IM for child’s development - or if “parental willingness for IM” reflects this argument than the sentence needs rewording. (consistency of terms in the whole manuscript) . Additionally the conclusion can be strengthen by some more informations about the novelty of this study and future research steps to take, e.g. in which way can the results be useful for interventions?
Another comment is to please work on some grammatical and formatting errors throughout the paper.
Some more specific comments with specific line numbers follow below:
Title (line 2-4): “children’s autonomy perception”
This wording implicit that children were asked how they perceive their autonomy. As only parents rated children’s autonomy the title needs rewording to avoid this misunderstanding.
Line 3: Please change “Predictor Factors” to “Predictors”
Line 25 and throughout the whole manuscript: The terms used for independent mobility vary throughout the whole manuscript: independent mobility to school (line 17), children’s independent mobility (line 110), go to school autonomously (line 25), autonomy (Table 2); abbreviations: IM (line 17), CIM (Table 3 & 4). Using one term e.g. “independent mobility (IM)” and “independent” consistently would help the reader to better understand the study. As additionally “child’s autonomy” is used as another factor associated with IM (line 165). (consistency of terms in the whole manuscript). In line with this, the used terms (no autonomy, one way and both ways) should be changed. My suggestion: no IM, IM one way, IM both ways
Line 30: Please change “autonomous perception” to “perception of autonomy” or think about another keyword.
Line 36 Please change “about” to “around”
Lines 49-51 “As demonstrated by Prezza et al. [15], it is increasingly more difficult to watch children playing in or walking along streets, open spaces, parks, etc., in western countries without strict adult supervision.” This sentence needs rewording.
Line 110: Please add a ‘s after children: children’s independent mobility
Line 119-120 and Table 2: Please use “number of children” or “number of offspring” consistently. (consistency of terms in the whole manuscript)
Line 129: Please add Spain after “Huesca”
Line 137: “years 4,5 and 6”
As school systems and class terms differ between countries please add an age range to enable a comparison with other studies.
Line 191: “which did away with the need to control the gender variable for the following analyses.” This sentence needs rewording.
Lines 142-148 and Table 1: The detailed description of the sample size is normally presented in the results section.
Table 2: What do “a” and “b” after the SD refer to? Please add this information in the table legend.
Table 4: Please change “Tabla” to “Table”
Line 288: In line with Bauman, Sallis, Dzewaltowski, and Owen (2002) please change “determinants” to “correlates”.
References
Bauman, A. E., Sallis, J. F., Dzewaltowski, D. A., & Owen, N. (2002). Toward a better understanding of the influences on physical activity: the role of determinants, correlates, causal variables, mediators, moderators, and confounders. Am. J. Prev. Med., 23(2 Suppl), 5-14. doi:10.1016/S0749-3797(02)00469-5
Buliung, R. N., Larsen, K., Faulkner, G., & Ross, T. (2017). Children’s independent mobility in the City of Toronto, Canada. Travel Behav. Soc., 9, 58-69. doi:10.1016/j.tbs.2017.06.001
Pacilli, M. G., Giovannelli, I., Prezza, M., & Augimeri, M. L. (2013). Children and the public realm: antecedents and consequences of independent mobility in a group of 11–13-year-old Italian children. Child. Geogr., 11(4), 377-393. doi:10.1080/14733285.2013.812277
Scheiner, J., Huber, O., & Lohmüller, S. (2019). Children's independent travel to and from primary school: Evidence from a suburban town in Germany. Transportation Research Part A: Policy and Practice, 120, 116-131. doi:https://doi.org/10.1016/j.tra.2018.12.016
Shaw, B., Bicket, M., Elliott, B., Fagan-Watson, B., Mocca, E., & Hillman, M. (2015). Children's Independent Mobility: an international comparison and recommendation for action. Policy Studies Institute London.
Author Response
RESPONSES TO REVIEWER 2
First of all, we would like to thank the reviewer for these comments. We also thank him/her for all the provided feedback as we believe that responding to the revision has improved the original version of the manuscript. We appreciate the constructive and thoughtful feedback and suggestions. We systematically describe the changes made to address the reviewer's suggestions, and we provide an account of each response.
Thank you for the opportunity to review this manuscript. The study described addresses an interesting and worthwhile topic – parental factors associated with different forms of children’s independent mobility to school. The study is well conceptualized and provides an in-depth view on the parental perspective of children’s autonomy.
In broad terms, the manuscript conforms to an appropriate structure and style, but at a more detailed level, it needs some work. There are a few matters of general grammatical construction and vocabulary. Furthermore, the use of different variants of essentially the same term - which may be deliberate and intended to make the narrative more varied - in my view just leads to more work for the reader in cross-referencing and confirming interpretations. It would be better to decide on a specific term and then using it consistently. There are also some issues regarding methodological explanations and analysis. Below I have made detailed suggestions for improvements.
Answer: We thank the reviewer for these suggestions. Following his/her suggestion, we have revised the whole manuscript and have made some changes in grammatical constructions and vocabulary. We also agree with the reviewer that the use of different variants of essentially the same term can mean more work for readers. Although our intention here was to make the narrative more varied, we realize that this could actually be a difficultly for readers. We sincerely thank the reviewer for his/her suggestion, and we have decided to use the same specific term throughout the manuscript. We have revised the whole manuscript to use only the word IM (independent mobility) throughout.
A major methodological concern is the missing inclusion of important child’s characteristics in the analysis (only number of children and position). Many studies evaluated that a child’s gender and age hardly influence the extent of independent mobility parents granted their child (Buliung, Larsen, Faulkner, & Ross, 2017; Pacilli, Giovannelli, Prezza, & Augimeri, 2013; Scheiner, Huber, & Lohmüller, 2019; Shaw et al., 2015). For example, Scheiner et al. (2019) found that the feeling that the child is too young is mainly associated with children younger than eight years.
Including only parents characteristics and opinions and including parents of children from three different school years (4-6) in the sample, can lead to a misinterpretation of the results, e.g. if all children of the “no autonomy” group are within the same age. For this reason, if age and gender of the child is available, the data should be analyzed again be separating age/gender groups or controlling for child’s age/gender.
Answer: We agree with this suggestion as both variables are undoubtedly very important. We analyzed these variables after taking this comment into account. Regarding children´s gender, we did not find any significant differences between boys and girls in their IM level [F (1, 1352) = .312, p = .517]. Nevertheless, in order to make age-based comparisons, we initially considered to group the children into three groups: 9-, 10-, and 11-year-oldd (only 4 children were 8 years old, and only 86 were 12 years old). However, we realized that these age groups would correspond to the school years which they belong to. So we decided to make school year-based comparisons, which are inherently related to children´s age, in an attempt to avoid redundancy by including both children’s age and school year. When this variable was considered, some differences were found for one of the regression analyses. Therefore, these data have been included in the analysis and the interpretation of our findings. If, however, the reviewer considers it necessary to also add this age variable, we will be willing to reanalyze and integrate this variable if necessary.
We have included this information in the revised manuscript, and have considered this variable in subsequent analyses (Table 2 and regression analyses).
Lines 243-247: Regarding children´s gender, no significant differences between boys and girls in their IM levels were found [F(1, 1352) = .312, p = .517]. However, significant differences in children’s IM according to school year were observed [F(2, 1349) = 91.75, p < .001] in all three years. Greater IM was found for the children in Primary Education year 6. Therefore, we included this variable in subsequent analyses.
Please add a section describing the applied statistical analysis of the study as last chapter of “Materials and Methods” to provide the opportunity of reproducibility. (after line 187)
Answer: Following this suggestion, a “data analysis” section has been included in the revised manuscript before the “Results” section as the last chapter of the “Materials and Methods” (lines 202-229).
In the conclusion (line 381) it is mentioned that the family value the encouragement of autonomy for their child’s development. However, no method or result is described which evaluated the importance of IM for child’s development - or if “parental willingness for IM” reflects this argument than the sentence needs rewording. (consistency of terms in the whole manuscript). Additionally the conclusion can be strengthen by some more informations about the novelty of this study and future research steps to take, e.g. in which way can the results be useful for interventions?
Answer: We agree with the reviewer that no method or result in this study that evaluates the importance of IM for child’s development is described. We believe that this phrase may have resulted from the confusion of terms employed in the previous version of the manuscript. So thank the reviewer for his/her valuable comment. Taking into account his/her feedback, we have eliminated this phrase and have also rewritten some sentences in the Conclusions. By following his/her suggestions, we have improved the Conclusions with more information about the novelty of this study and future research steps.
Lines 422-441: In recent decades, children’s IM to school has substantially reduced, which implies negative consequences for their health and psychosocial well-being. This study analyzed several family variables associated with different forms (one way and both ways) of children’s IM to school, and it provides interesting and novel data about these variables. The obtained results reveal that the families that permit more IM (both ways) are those that wish their children to go to school alone, perceive fewer difficulties in their social setting, and consider their children more autonomous. As the results of this study highlight the importance of parental willingness for greater IM, it would be convenient to continue exploring parents' reasons for such willingness and to what extent their perception of the importance of enhancing children’s autonomy may be related with allowing their children’s IM to school. One of the main contributions of this study is the importance of parental willingness for children’s IM to school, and future studies should profoundly examine this variable and which factors influence it.
Taking into account the importance of parental perceptions, it would also be advisable to develop interventions to make parents aware of the benefits that children’s IM to the school has for the health and psychosocial well-being of their children. Although some environmental and organizational aspects of cities (e.g., accessible streets, fewer cars, distance to school, etc.) should be changed to favor children’s IM to school, parental perception of the possible dangers and difficulties for children’s IM, and probably what they think about the behaviors that are also socially expected of a “good father”/ “good mother”, could also be key variables to help increase children’s IM to school in our societies.
Another comment is to please work on some grammatical and formatting errors throughout the paper.
Answer: Following this suggestion, we have revised the whole manuscript and have corrected some grammar and formatting mistakes.
Some more specific comments with specific line numbers follow below:
Title (line 2-4): “children’s autonomy perception”
This wording implicit that children were asked how they perceive their autonomy. As only parents rated children’s autonomy the title needs rewording to avoid this misunderstanding.
Line 3: Please change “Predictor Factors” to “Predictors”
Answer: We thank the reviewer for these comments. Following his/her suggestions, we have changed the manuscript title.
Lines 2-4: Parents´ Willingness and Perception about Children’s Autonomy as Predictors of Greater Independent Mobility to School.
Line 25 and throughout the whole manuscript: The terms used for independent mobility vary throughout the whole manuscript: independent mobility to school (line 17), children’s independent mobility (line 110), go to school autonomously (line 25), autonomy (Table 2); abbreviations: IM (line 17), CIM (Table 3 & 4). Using one term e.g. “independent mobility (IM)” and “independent” consistently would help the reader to better understand the study. As additionally “child’s autonomy” is used as another factor associated with IM (line 165). (consistency of terms in the whole manuscript)
Answer: We agree with the reviewer about the confusion that using different terms can lead to. We have reviewed the whole manuscript and used the term IM (independent mobility) throughout.
In line with this, the used terms (no autonomy, one way and both ways) should be changed. My suggestion: no IM, IM one way, IM both ways
Answer: Following this suggestion, in the whole revised manuscript we have used: no IM, IM one way, IM both ways. We believe that these terms are clearer and avoid confusion between the autonomy and independent mobility concepts (which coincides with what another reviewer indicated). We thank the reviewer for his/her suggestion.
Line 30: Please change “autonomous perception” to “perception of autonomy” or think about another keyword.
Answer: Following this suggestion, we have changed “autonomous perception” to “perception of autonomy” (line 30)
Line 36 Please change “about” to “around”
Answer: Following this suggestion, we have changed “about” to “around” (line 36).
Lines 49-51 “As demonstrated by Prezza et al. [15], it is increasingly more difficult to watch children playing in or walking along streets, open spaces, parks, etc., in western countries without strict adult supervision.” This sentence needs rewording.
Answer: Following this suggestion, we have rewording this sentence (lines 57-59): According to Prezza et al. [22], watching children playing or walking in streets, open spaces, parks, etc., in western countries without strict adult supervision is increasingly more difficult.
Line 110: Please add a ‘s after children: children’s independent mobility
Answer: In the revised manuscript, this sentence has been rewritten (lines 128-129): These children also perceive their trip to school as being safe, unlike the children who do not practice IM to school.
Line 119-120 and Table 2: Please use “number of children” or “number of offspring” consistently. (consistency of terms in the whole manuscript)
Answer: We thank the reviewer for this comment. In the revised manuscript, we have changed “number of offspring” to “number of children” (lines 137-138).
Line 129: Please add Spain after “Huesca”
Answer: We have added Spain after “Huesca” (line 148).
Line 137: “years 4,5 and 6”
As school systems and class terms differ between countries please add an age range to enable a comparison with other studies.
Answer: We have added an age range (line 156): ….,which usually includes ages from 9 to 11 years old.
Line 191: “which did away with the need to control the gender variable for the following analyses.” This sentence needs rewording.
Answer: This sentence has been reworded as follows: “As no significant differences were found, further control of parents´ gender was not considered.” This sentence can be found in the “data analysis” section (lines 204-205).
Lines 142-148 and Table 1: The detailed description of the sample size is normally presented in the results section.
Answer: Following this suggestion, the detailed description of the sample and Table 1 have been included in the Results section (lines 231-237).
Table 2: What do “a” and “b” after the SD refer to? Please add this information in the table legend.
Answer: We thank the reviewer for this comment because we realized that we had used the subscripts in Table 2 unclearly, which could confuse readers. In the original version of the manuscript, the same subscripts indicated significant differences between the means of different groups. However in the revised version of the manuscript, we have used the subscripts in the that they are normally used; that is, different subscripts indicate significant differences between means. In the revised manuscript, we have changed some subscripts in Table 2 to provide this information more clearly. We have also placed a note at the end of the table to explain the meaning of the subscripts.
Line 275:
Note. Different subscripts indicate significant differences between means (p < .05): a < b < c
Table 4: Please change “Tabla” to “Table”
Answer: We have changed “Tabla” to “Table” (Table 4).
Line 288: In line with Bauman, Sallis, Dzewaltowski, and Owen (2002) please change “determinants” to “correlates”.
Answer: We thank the reviewer for this comment, and we have changed “determinants” to “correlates” (line 329).
References
Bauman, A. E., Sallis, J. F., Dzewaltowski, D. A., & Owen, N. (2002). Toward a better understanding of the influences on physical activity: the role of determinants, correlates, causal variables, mediators, moderators, and confounders. Am. J. Prev. Med., 23(2 Suppl), 5-14. doi:10.1016/S0749-3797(02)00469-5
Buliung, R. N., Larsen, K., Faulkner, G., & Ross, T. (2017). Children’s independent mobility in the City of Toronto, Canada. Travel Behav. Soc., 9, 58-69. doi:10.1016/j.tbs.2017.06.001
Pacilli, M. G., Giovannelli, I., Prezza, M., & Augimeri, M. L. (2013). Children and the public realm: antecedents and consequences of independent mobility in a group of 11–13-year-old Italian children. Child. Geogr., 11(4), 377-393. doi:10.1080/14733285.2013.812277
Scheiner, J., Huber, O., & Lohmüller, S. (2019). Children's independent travel to and from primary school: Evidence from a suburban town in Germany. Transportation Research Part A: Policy and Practice, 120, 116-131. doi:https://doi.org/10.1016/j.tra.2018.12.016
Shaw, B., Bicket, M., Elliott, B., Fagan-Watson, B., Mocca, E., & Hillman, M. (2015). Children's Independent Mobility: an international comparison and recommendation for action. Policy Studies Institute London.
Answer: We thank the reviewer very much for his/her time and dedication in providing the references. These references have been included as they have helped enrich the theoretical framework, support our findings, and improve our manuscript.

Reviewer 3 Report
Dear Authors,
the theme was very interesting and the research was worthy. I appreciated the richness of the study and the variables considered. However, I have a few remarks that I would like to consider before arriving at the publication.
1. Autonomy and independence are treated as similar concept and construct, whereas the first one in the literature is used with a broader significance. I think that authors should justify their decision, introducing a paragraph that explain why they assimilate autonomy with self-care or personal independence.
2. line 165-166; dependence vs independence (autonomy is something different)
3. lines 118-126 and lines 202-205. I think that some socio-demographic variables like number of children in the family and offspring position could not be considered as dependent variables, but as independent variables influencing the children's IM. Statistical analyses are not appropriate. I would suggest log linear analyses to test association (or independence) between three or more categorical variables.
4. Authors should also consider possible influences of cultural perception of autonomy and independence in promoting child behaviour. Following this line I suggest to discuss their results in their results. I would suggest I suggest discussing their results in light of the contribution offered by the following articles:
a. Taverna, L., Bornstein, M.H., Putnich, D.L., Axia, G. (2011). Adaptive Behaviors in young Children: A Unique Cultural Comparison in Italy. Journal of Cross Cultural Psychology, 42(3), 445–465. doi: 10.1177/0022022110362748.
b. Taverna, L., Tremolada, M., Bonichini, S. (2017). Conoscenze materne e sviluppo del bambino in due gruppi culturali altoatesini. Ricerche di Psicologia, 40(2), 257-278. ISSN: 0391-6081. ISSNe: 1972-5620. DOI: 10.3280/RIP2017- 002005.
Best regards.
Author Response
RESPONSES TO REVIEWER 3
First of all, we would like to thank the reviewer for these comments. We also thank him/her for all the provided feedback as we believe that responding to the revision has improved the original version of the manuscript. We appreciate the constructive and thoughtful feedback and suggestions. We systematically describe the changes made to address the reviewer's suggestions, and we provide an account of each response.
Dear Authors,
the theme was very interesting and the research was worthy. I appreciated the richness of the study and the variables considered. However, I have a few remarks that I would like to consider before arriving at the publication.
1. Autonomy and independence are treated as similar concept and construct, whereas the first one in the literature is used with a broader significance. I think that authors should justify their decision, introducing a paragraph that explain why they assimilate autonomy with self-care or personal independence.
Answer: We thank the reviewer for this suggestion and we agree that autonomy and independence were incorrectly treated as a similar concept and construct in the original manuscript. Certainly, autonomy has a much broader meaning when used in the literature. So we believe that the original manuscript was confusing for readers. After taking into account this comment, we have thoroughly reviewed the whole manuscript. In the revised version, we have used the independence concept properly and have clarified the autonomy concept in the Introduction.
Lines 87-91: The autonomy concept is defined as a state of being independent or self-governing, and frequently refers to three domains: behavioral, emotional and cognitive [33]. Parents who perceive their children as being more autonomously in cognitive, emotional and behavioral terms could also be more favorable to allow them greater IM. Moreover, parents’ attitude about the importance of enhancing their children’s autonomy can be a relevant variable.
2. line 165-166; dependence vs independence (autonomy is something different)
Answer: We agree with the reviewer that independence and autonomy are related, but are not the same construct. After taking into account this comment, we have made a change in the revised manuscript (line 180): … for which there were two options: not autonomous vs. autonomous.
3. lines 118-126 and lines 202-205. I think that some socio-demographic variables like number of children in the family and offspring position could not be considered as dependent variables, but as independent variables influencing the children's IM. Statistical analyses are not appropriate. I would suggest log linear analyses to test association (or independence) between three or more categorical variables.
Answer: We agree with the reviewer that the socio-demographic variables could not be considered to be dependent variables, but as independent variables that influence children’s IM. We considered IM in this manner in the original manuscript, but our description of the performed analyses was probably confusing. In lines 237-244 (in the original manuscript) we indicated: Tables 3 and 4 show the binary logistic regression analysis results for which a dependent variable was established in each case: a) no autonomy vs. one-way (the dichotomous variable was 0 and 1, respectively) and b) no autonomy vs. both ways (the analyzed dichotomous variable was 0 and 2, respectively).
Certainly, we believe that the description of the analyses in the original manuscript was confusing and unclear. Specifically, our description of the MANOVA in the original manuscript could easily confuse readers. We used MANOVA to compare any possible differences among the three groups with different children’s IM. In the description of these analyses, we wished to indicate that IM was the variable of contrast (in the SPSS program: factor or independent variable), but only for MANOVA, which does not mean that we considered IM to be the independent variable to predict the other variables examined in the present study. We really believe that the previous description of the analyses was confusing, and we thank the reviewer who helped us by making his/her comment to improve it. After taking it into account, we have included a clearer description of the analyses performed in this study in the revised manuscript.
Lines 202-229: 3. Data analysis
First of all, the sample’s socio-demographic characteristics were analyzed. A comparative analysis was done by performing a MANOVA of the variables evaluated for mothers and fathers. As no significant differences were found, further control of parents´ gender was not considered. We next examined the percentage of parents who indicated that their children went to school with no adult accompanying them (IM to school) by distinguishing: who went to school accompanied by adults (no IM); who went to or returned from school alone (IM one way); who went to and returned from school alone (IM both ways). At this point, we also examined whether there were differences in IM based on children´s gender and school year.
We analyzed if there were any significant differences in children’s IM level by considering the socio-demographic and parents´ perception variables. To do so, we conducted a MANOVA to analyze any possible differences among the three groups with different levels of children’s IM (no IM, IM one way, IM both way) in the socio-demographic variables (children’s school year, number of children, their position in the family, living with either both parents or only one, and parents’ age, nationality, level of education and job status) and the variables related to parents’ perceptions and attitudes (parents´ perceptions about their children’s autonomy, perceived difficulty of IM one way (to school), the reasons for not allowing IM to school, parental willingness for their childten’s IM to school, giving their child house keys, frequency of IM for other leisure activities and perceived dangers). In these analyses, the categorical variables, which were mostly dichotomous, were treated as continuous variables. As there were three groups with different levels of children’s IM, a post hoc test were run to explore the significant differences between means.
Finally, in order to analyze the predictive value of the examined variables (socio-demographic variables and the variables related to parents’ perceptions and attitudes) on children’s IM, we conducted several binary logistic regression analyses, in which a dependent variable was established in each case as follows: a) no IM vs. IM one-way (the dichotomous variable was 0 and 1, respectively) and b) no IM vs. IM both ways (the dichotomous variable was 0 and 2, respectively). The socio-demographic variables and variables related to parents’ perceptions and attitudes were included in the regression analyses as predictor (independent) variables.
Regarding the analyses done to explore the predictive capacity of the socio-demographic variables and variables related to parents’ perceptions and attitudes, we have considered that dependent variable (IM) is categorical, as indicated by the reviewer. So we have conducted a logistic regression analysis, which is adequate when the dependent variable is categorical.
4. Authors should also consider possible influences of cultural perception of autonomy and independence in promoting child behaviour. Following this line I suggest to discuss their results in their results. I would suggest I suggest discussing their results in light of the contribution offered by the following articles:
a. Taverna, L., Bornstein, M.H., Putnich, D.L., Axia, G. (2011). Adaptive Behaviors in young Children: A Unique Cultural Comparison in Italy. Journal of Cross Cultural Psychology, 42(3), 445–465. doi: 10.1177/0022022110362748.
b. Taverna, L., Tremolada, M., Bonichini, S. (2017). Conoscenze materne e sviluppo del bambino in due gruppi culturali altoatesini. Ricerche di Psicologia, 40(2), 257-278. ISSN: 0391-6081. ISSNe: 1972-5620. DOI: 10.3280/RIP2017- 002005.
Answer: We thank the reviewer for his/her suggestion and for providing the references. These references have been useful for improving the discussion of the results in the revised manuscript. We have included them all in our revised manuscript.
Lines 407-412: In addition, cultural aspects linked to parental socialization styles and social norms should also be considered in future research. In this line, previous studies [58,59] have shown that cultural differences are related to differences in parental childrearing strategies and parental socialization goals. Taverna et al. [58] observed differences between German and Italian cultures in childrearing practices, and the levels of autonomy and independence that mothers expect in their children.
Finally, we would like to thank the anonymous reviewer for all his/her comments, which have enabled us to improve our manuscript.
Round 2
Reviewer 2 Report
Dear authors,
thank you for the reviesd version of the manuscript, which has been clearly improved.
Just one little grammatical comment considering the title of the manuscript:It must be 'perception of children's automomy".
Reviewer 3 Report
You have improved your study. Good work.